# Are Spine-Bearing Freshwater Gastropods Better Defended?

**Andrew R. Davis [1,\*], Matthew J. Rees [1], Bindiya Rashni [2] and Alison Haynes [2,†]**

1   Centre for Sustainable Ecosystem Solutions & School of Biological Sciences, University of Wollongong, Wollongong, NSW 2522, Australia; mjr849@uowmail.edu.au
2   Institute of Applied Sciences, University of the South Pacific, Suva, Fiji; diyarash@gmail.com
\*   Correspondence: adavis@uow.edu.au; Tel.: +61-2-4221-3432
†   The author passed away on 27 February 2016.

**Abstract:** Freshwater snails usually possess thin unadorned shells lacking structural components such as spines. Exceptions can be found on the high, well-watered islands of the South Pacific. Streams on these islands support a rich freshwater molluscan fauna with several nerite taxa (Neritimorpha: Neritidae) exhibiting extremely long dorsal spines. We sought to assess the defensive capacity of these structures for several co-occurring nerite genera on the Island of Ovalau, Fiji. Our overarching hypothesis was that spines confer a defensive advantage. We tested four predictions for eight common taxa: (i) predator "rich" habitats (the creek entrance) would be dominated by spine-bearing nerites, (ii) spine-bearing species should be smaller in size, (iii) nerites with spines would exhibit lower levels of shell damage and (iv) nerites with spines should invest less in their shells (i.e., their shells should be thinner). Most of these predictions received support. Spine-bearing species dominated the entrance to the creek and were smaller in size. Levels of shell damage were low overall, with 2 of the 3 spinose taxa exhibiting no shell damage, as did many of the nonspinose taxa. Finally, shells of spinose species were 25% thicker, demonstrating increased rather than decreased investment. Taken together, these findings suggest that the elaborate spines of *Clithon* spp. play a defensive role.

**Keywords:** Ovalau Island; Fiji; *Clithon*; *Neritina*; *Neritona*; defense

## 1. Introduction

Predation can have pervasive effects on the ecology and evolution of prey and predators alike [1,2]. As a defensive strategy, there is a wealth of evidence that spines play an important protective role against predators in both terrestrial and aquatic systems [3,4]. Among invertebrates, the molluscs have shown overwhelming evidence that shell thickening and ornamentation, such as the presence of ribs, knobs or spines, act to slow or disrupt the actions of predators [5–8]. Shell ornamentation such as spines may not act wholly to directly impair predators though. The freshwater snail, *Tiphobia horei*, in Lake Tanganyika, Eastern Africa, is purported to use its elongate spines to reduce sinking in soft mud [9]. There is also evidence among spondylid bivalves that spines encourage the recruitment and growth of epibionts that then serve to camouflage their host, enhancing host survival [10].

Most evidence for the defensive role of molluscan spines comes from the marine environment and it is in the ocean that shell ornamentation is most impressive [11]. The argument has been that the ocean is a more predator "rich" environment [12] requiring enhanced defenses. It is widely acknowledged that freshwater molluscs lack the shell development and ornamentation of their marine counterparts [11,12]. The reasons are probably twofold—constrained by the availability of calcium carbonate and less pressure from predators in freshwater environments. There are some exceptions to the development of spines in freshwater taxa, among these are members of the tropical freshwater

nerite genus *Clithon.* These taxa often possess very long spines [12–14], exceeding the length of the shell in some instances.

Nerites are an exceptional group of molluscs—the Neritimorpha (formerly Neritopsina) forms a distinct clade of marine, terrestrial and freshwater taxa containing several hundred species [15,16]. This group has successfully invaded freshwater habitats on numerous occasions [15]. In the tropical South Pacific, many high islands have developed a rich nerite stream fauna with some 33 species drawn from five genera [17,18]. The islands of Fiji, for example, host more than 23 stream nerite taxa. The freshwater nerites are also unusual in possessing an amphidromous life history where adults live and breed in freshwater, while larvae are swept to the ocean and usually undergo an extended marine dispersive phase [19,20]: a life history characteristic also seen in some fishes and crustaceans [21,22]. Among nerites, settling juveniles then return to freshwater and crawl upstream, with some taxa even "hitchhiking" on the shells of congeners [23,24].

The streams of the high islands of Fiji support nerite fauna bearing spines and those lacking them—a perfect natural laboratory in which to contrast the defensive role of spines. Our overarching null hypothesis was that spines possess no defensive advantages, and we tested this for eight common taxa, three spinose *Clithon* spp. and five taxa lacking spines; another *Clithon* species., a species of *Neritona* and three *Neritina* spp. We sampled at several stations in Rukuruku creek on the Island of Ovalau to test four predictions relating to the defensive role of spines: (i) predator "rich" habitats (the creek entrance) would be dominated by spine-bearing nerites; (ii) spines would advantage diminutive taxa, and so spine-bearing species should be smaller in size; (iii) nerites with spines would exhibit lower levels of shell damage; (iv) nerites with spines should invest less in their shells (i.e., their shells should be thinner). Most of these predictions were borne out, forcing us to reject our null hypothesis and conclude that spines do indeed play a defensive role in spinose members of the genus *Clithon*.

## 2. Materials and Methods

### 2.1. Study Location and Sampling

Our focus was on several stations on a single creek on the Island of Ovalau, east of the capital Suva (Figure 1). Rukuruku Creek (17°39′0″ S 178°45′59.99″ E) is a relatively small, fast flowing creek characteristic of many in the higher, wetter islands of the Fijian archipelago (author's personal observation). We sampled at 4 stations along Rukuruku Creek, 100, 600, 1000 and 2000 m from the creek entrance. The lower three stations were amidst native gardens with little tree cover. The topography at these lower stations was relatively flat, the creek shallow (<0.25 m in depth) and with flow rates of around 10 cm/s per second. The 2000 m station was in a steeper, boulder strewn area of the creek and included some large pools. At this station, the stream was forested, flowing rapidly (>20 cm/s per second) and was rarely more than 2.5 m wide. We contend that the fauna of Rukuruku Creek is typical of other creeks on the high islands of the Fijian archipelago, with many of the taxa we examined widely distributed across the tropical South Pacific [17,18].

Each station was searched for freshwater molluscs for approximately 1 h on each of two occasions. Searches included the overturning of boulders and snorkeling in some of the deeper pools. Most molluscs were identified, measured and released back into their appropriate habitat. For each mollusc, we measured shell length, width and height using vernier calipers, and shells were inspected for damage or signs of repair. Spines, where present, were counted and measured using vernier calipers. A voucher sample was returned to the laboratory to confirm identification and for determinations of shell thickness (see below). Identification was confirmed with the aid of a microscope at the laboratory at the University of the South Pacific, Suva campus.

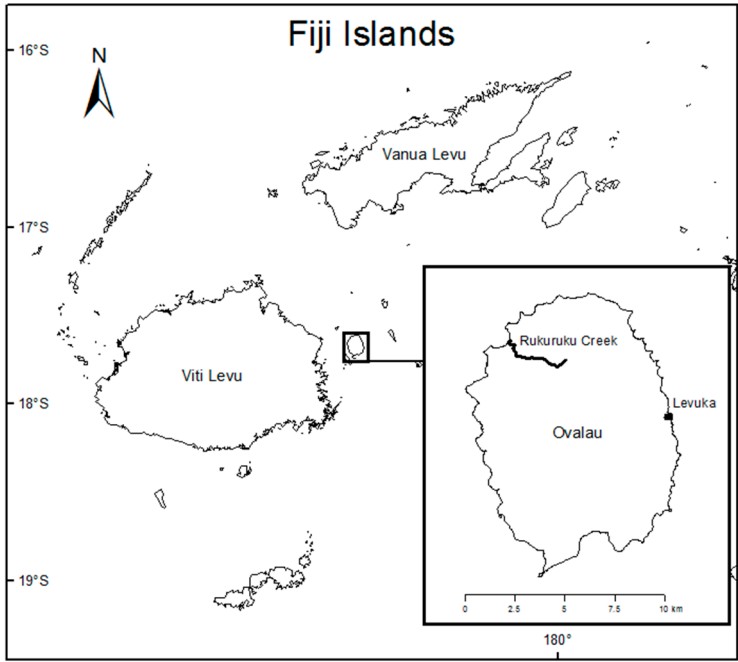

**Figure 1.** Study location, Rukuruku Creek, Island of Ovalau, Fiji.

Shell thickness was determined from images acquired with a Leica microscope (M216A). Shells that had been fixed in alcohol (70%) were cut down the dorsal axis with a dremel handheld saw with a diamond cutting disk. This yielded two halves, both of which were photographed and processed in the "Leica Application Suite". For each shell half, five points were demarcated and shell thickness measured, yielding a total of 10 shell-thickness measures for each individual.

*2.2. Data Analysis*

To test for differences in shell thickness between spinose and nonspinose nerites, we fitted linear mixed-effects models. Mixed-effects models were chosen over simple linear models to account for hierarchical groupings and repeated measures [25]. To test whether shell thickness differed between spinose and nonspinose nerites across all fauna, "taxa" and "individual" (nested within "taxa") were included as random factors. To test for differences in shell thickness between spinose and nonspinose individuals of *Clithon pritchardi*, "individual" was included as a random factor. In both models, the shell length of each individual was included as a fixed covariate, as shell thickness was expected to correlate positively with length.

Linear mixed-effects models were fitted using the "lmer" function from the "lme4" package [26] in the statistical software, R (Version 3.6.3, Vienna, Austria) [27]. The "lmerTest" package was used to determine F and *p* values based on Satterthwaite's method, to test the statistical significance of spine presence on shell thickness [28]. Plots of the relationship between shell thickness and shell length between nonspinose nerites were generated using the "ggplot2" package [29] and "visreg" package [30]. The "ggeffects" package was used to obtain and plot 95% confidence intervals [31].

**3. Results**

As expected, we encountered a highly diverse fauna of freshwater molluscs on Rukuruku creek, with at least 15 nerite taxa and four species of Thiarids, despite the modest size of the creek (Appendix A). Our focus was on the members of the genera *Clithon*, *Neritina* and *Neritona*. *Clithon* and *Neritina* were each represented by four species, while *Neritona* was represented by a single taxon. Three of the *Clithon* spp. bore large spines (Figures 2 and 3), while spines were absent within *Neritina* and *Neritona*.

Two forms of *C. pritchardi* were encountered: one spinose and the other lacking spines. In the case of *C. diadema*, spines averaged more than 40% of the maximum shell length of individuals.

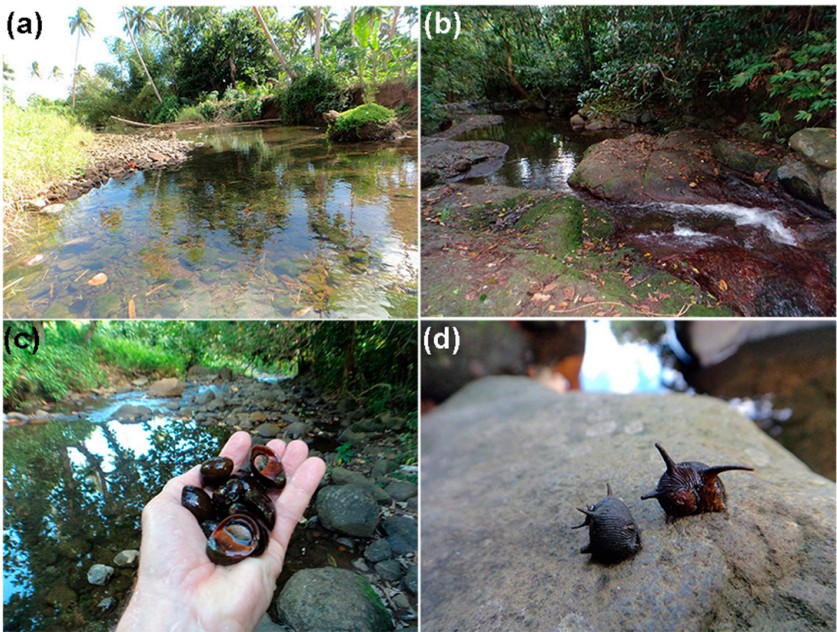

**Figure 2.** Typical modest-sized Pacific high island creek and representatives of nerite fauna, Rukuruku Creek, Island of Ovalau, Fiji. (**a**) Pool at creek entrance dominated by *Clithon corona* and *C. diadema*. (**b**) Pool ≈ 2 km above the entrance looking downstream, dominated by *C. olivaceum*. (**c**) *Neritina petitii*, (**d**) *Clithon pritchardi* (spinose form).

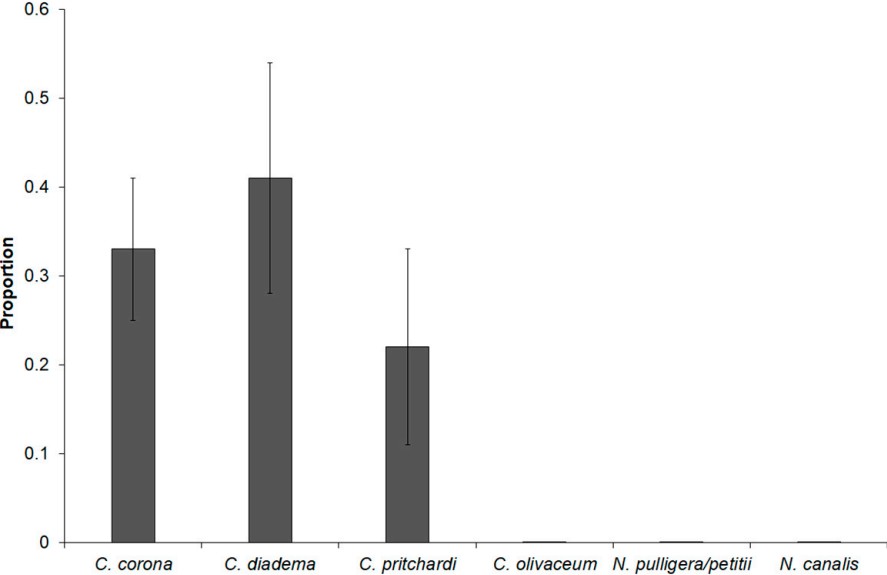

**Figure 3.** Average spine length as a proportion of total shell length for the six common nerite taxa sampled from Rukuruku Creek on the Island of Ovalau, Fiji. Error bars are the standard deviation and sample sizes as outlined in Table 1.

**Table 1.** Relative abundance of spinose and nonspinose nerite taxa in the genera *Clithon* (*C.*), *Neritina* (*N.*), *Neritona* (*Ne.*), with increasing distance from the entrance to Rukuruku Creek, Ovalau Island, Fiji.

| | Distance from Entrance (m) | | | |
|---|:---:|:---:|:---:|:---:|
| | **100** | **600** | **1000** | **2000** |
| *C. corona* | ● | | | |
| *C. diadema* | ● | | | |
| *C. pritchardi* | ● | ● | ● | |
| *C. pritchardi* * | | ○ | ○ | |
| *C. olivaceum* | | ○ | ○ | ○ |
| *N. pulligera/petitii* | ○ | ○ | ○ | ○ |
| *N. porcata* | | ○ | ○ | |
| *N. canalis* | | | ○ | ○ |
| *Ne. macgillivrayi* | | | ○ | |

\* Nonspinose form ● ≤ 0.10, ● = 0.10–0.25, ● ≥ 0.26; ● = spinose individuals; ○ = nonspinose individuals.

Our initial prediction that the "predator-rich" lower reaches of the creek would be dominated by spinose gastropods was borne out. At the station closest to the stream entrance (100 m), more than 0.95 of the sample comprised spinose individuals (Figure 4). This proportion dropped markedly as we moved further up the creek, and at 2000 m beyond the entrance, no spinose snails were observed (Figure 4). *Clithon corona* and *C. diadema* dominated the sample at the creek entrance, with a number of spinose *C. pritchardi* also collected (Table 1). The only unspined species near the entrance was *Neritina pulligera* or *N. petitii* (these two taxa were not distinguished in the field). Spinose *C. pritchardi* were still well represented at our 600 m sample station as was the unspined form of this taxon. At the 1000 m sample station, spinose taxa no longer dominated the sample; the spined form of *C. pritchardi* was evident, but several unspined *Neritina* spp. now dominated, particularly *N. pulligera/petitii*. At the 2000 m sample station, no spinose taxa were observed, and *C. olivaceum* dominated the sample.

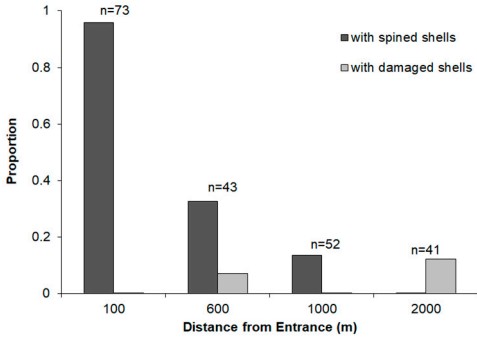

**Figure 4.** Proportion of sample, pooled across all taxa, bearing spines and proportion showing evidence of shell damage with increasing distance from the entrance to Rukuruku Creek, Ovalau Island, Fiji.

Our second prediction was that spinose species should be smaller in size, and this was borne out for *Clithon corona* and *C. diadema*. These were the most diminutive taxa we encountered (Table 2).

The smallest individuals of *C. pritchardi* bore spines, but there was considerable overlap in size with the nonspinose form of this species. The most robust taxon we encountered was *N. pulligera/petitii* (Figure 2, Table 2).

**Table 2.** Shell features for freshwater nerite taxa, Rukuruku Creek, Fiji. Maximum shell length (SL) from Eichhorst [14]. Sample sizes for *Neritina porcata* and *Neritona macgillivrayi* were deemed too small for inclusion.

| Species | Shell Length (Max SL) mm | Max. No. of Spines | Max. Length of Spines | % Shells Damaged | n |
|---|---|---|---|---|---|
| *Clithon corona* | 7.9–16.9 (27) | 6 | 5.0 mm | 0 | 17 |
| *Clithon diadema* | 3.8–10.4 (16) | 4 | 4.5 mm | 0 | 54 |
| *Clithon pritchardi* * | 9.8–22.0 (24) | 4 | 5.8 mm | 5.8 | 17 |
| *Clithon pritchardi* | 14.7–26.0 (24) | 0 | | 4.0 | 25 |
| *Clithon olivaceum* | 14.1–24.0 (40) | 0 | | 6.8 | 44 |
| *Neritina pulligera/petitii* | 17.0–37.0 (43) | 0 | | 6.4 | 31 |
| *Neritina canalis* | 20.1–26.0 (25) | 0 | | 0 | 8 |

* spinose form.

We also predicted that nerites with spines should enjoy lower levels of shell damage. We detected very low levels of shell damage overall—rarely more than 12% of a sample from any location (Figure 4). Our prediction found some support in that we failed to detect shell damage on the spinose *Clithon corona* or *C. diadema* (Table 2). However, the spinose form of *C. pritchardi* exhibited similar levels of damage to the unspined form at around 5% of the sample for that taxon. The robust *Neritina pulligera/petitii* showed evidence of shell damage, albeit quite low, while others did not (Table 2).

In contrast to our final prediction that spine bearing taxa should invest less in their shells, we found that when controlling for shell size, spinose taxa had shells that were 25% thicker than taxa lacking spines, equating to an average of 0.1 mm (Figure 5). A thicker shell was also apparent for the spinose forms of *Clithon pritchardi* when compared to the nonspinose members of this species (Figure 5)—effects that proved to be marginally nonsignificant (Table 3).

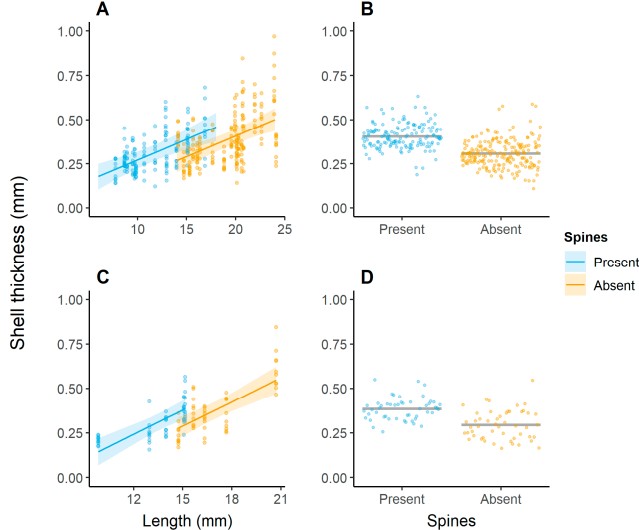

**Figure 5.** Relationships between shell thickness and shell length for spinose and nonspinose nerites within Rukuruku Creek, Ovalau Island, Fiji. Mean predictions from linear mixed effects models for shell thickness for (**A**) the entire fauna across shell length, (**B**) the entire fauna at mean shell length, (**C**) *Clithon pritchardi* across shell length and (**D**) *Clithon pritchardi* at mean shell length. Raw data points and 95% confidence intervals are presented in (**A,C**). Partial residuals from the linear mixed models are presented in (**B,D**).

**Table 3.** Linear mixed effects model analysis of shell thickness for: A, a range of spinose and nonspinose nerite taxa with species and individuals as nested random terms; B, *Clithon pritchardi* including individuals as a random factor.

| | Sum of Squares | Numerator DF | Denominator DF | *f* (Value) | *p* |
|---|---|---|---|---|---|
| **(A) All taxa** | | | | | |
| Spines | 0.023 | 1 | 9.14 | 3.75 | 0.084 |
| Length | 0.125 | 1 | 14.94 | 19.94 | **<0.001** |
| **(B) *C. pritchardi*** | | | | | |
| Spines | 0.022 | 1 | 6.99 | 3.84 | 0.091 |
| Length | 0.167 | 1 | 6.99 | 28.90 | **0.001** |

## 4. Discussion

Outcomes of the predictions that we tested for freshwater nerites in the tropical Pacific were consistent with predators having shaped the morphology and responses of this freshwater fauna. The spectacular elaboration of spines in *Clithon* spp. (Figures 2 and 3) support the notion of their defensive role, as most of our predictions were borne out. Of particular note was the domination of the "predator rich" entrance to the creek by the three species of spinose nerites observed in this study. Coevolutionary interactions among freshwater predators and their prey are not unprecedented. A suite of endemic freshwater gastropods responded to an endemic shell-crushing crab by increasing their size, shell strength and shell sculpture in Lake Tanganyika, Africa [9]. Gastropods can employ a wide array of defensive strategies, and these are not restricted to shell morphology [32]. Indeed, further support for the view that predators have shaped this Indo-Pacific fauna can be drawn from the additional antipredator adaptations they exhibit. Specifically, these nerite taxa were predominantly nocturnal, capable of rapid movement and produced copious mucus—all characteristics consistent with predator avoidance [11].

Elaborations of molluscan shells, such as spines and foliaceous varices, may act in several ways to diminish the effectiveness of shell crushing (durophagous) predators, such as crabs or fishes. The presence of spines increases the effective diameter of shells, with pufferfish (*Diodon* spp.) requiring larger gape sizes to be effective at crushing prey [7]. Similarly, the presence of foliaceous varices increased the work required to break shells (work to failure) by 50 to 100%, suggesting longer handling times [8]. Spines may also act to spread the crushing forces exerted by predators, reducing the likelihood of catastrophic shell failure [7]. The spines of *Clithon* spp. are quite unlike the stout spinose sculpture observed in many marine species as they are long, narrow and hollow. The length of these spines does significantly increase the effective diameter of the shell, perhaps as much as 40%, and this may be of particular importance to diminutive taxa, such as *C. corona* and *C. diadema* in the present study. Their hollow nature likely renders them more prone to breakage, increasing the likelihood of their lodging in the mouths of would-be predators.

Several studies have highlighted the marked spatial variation in predation pressure and the responses of shelled molluscs to the presence of predators, even at the scale of ocean basins. The tropical Indo-Pacific is considered "relatively richer in predators" [5], and molluscs in the region have a higher incidence of defensive shell architecture. Knobs and spines were present on 20–35% of Indo-Pacific marine gastropods compared with 10–18% of Atlantic taxa and just 2% of temperate species [33]. Crab predators in the Indo-Pacific are also able to more effectively crush prey than their Atlantic and eastern Pacific counterparts [34]. It follows then, that the entrance to creeks in the Indo-Pacific will be rich in durophagous predators and that molluscs in this habitat will respond to this elevated threat. Indeed, several families of invertebrate feeding fishes are well represented in this habitat [35,36] including the Crescent Grunter—*Terapon jarbua*—which we observed in pools at the entrance to Rukuruku Creek. Predators are not restricted to creek entrances, however. The mid and upper reaches of freshwater systems in Fiji are frequented by several invertebrate feeders including

eels *Anguilla* spp., the goby, *Awaous guamensis*, the gudgeon, *Eleotris fusca* and several representatives of the Kuhliidae (David Boseto, Ecological Solutions, Solomon Islands, pers. comm.) [37]. We wish to emphasize though that we have inferred an elevated predation risk at the creek entrance, but it was not quantified.

Further support for the antipredator function of spines and its link to habitat can be drawn from a broader assessment of the literature on spinose nerites. Eichhorst [14] presents habitat information for 53 valid *Clithon* species, 18 of which bear spines. We contrasted spine bearing and nonspinose species across two habitats—brackish and freshwater—to determine if the presence of spines was contingent on habitat. Almost 78% of spine-bearing *Clithon* spp. were found within a brackish water habitat (i.e., the predator-rich entrance of creeks), and the presence of spines was significantly contingent on habitat ($\chi^2 = 4.17$, d$f = 1$, $p = 0.041$).

An alternate explanation for the abundance of small, spine-bearing snails at creek entrances stems from the amphidromous life history of these freshwater gastropods. Amphidromy is where adults are resident in freshwater while their larvae are swept downstream for an extended planktotrophic existence in the ocean before re-entering freshwater as postlarvae. Freshwater nerites across the Indo-west Pacific are recognized for upstream migrations, sometimes of many km [38,39]. Spine breakage as animals move upstream may explain the abundance of spinose taxa near the entrance to creeks, whereas spines are lost further upstream; Haynes [17] invoked this explanation in spinose *Clithon pritchardi*. However, the two most spinose taxa we encountered, *Clithon corona* and *C. diadema*, were restricted to the lower reaches of Rukuruku Creek. In addition, the smallest, most spinose individuals of *C. pritchardi* possessed a relatively thickened shell—indicative of change in shell morphology near the creek entrance and lending support to the notion that changes are driven by pressure from predators.

It is usually assumed that investments in defense are costly [40] and entail tradeoffs [41]. We predicted that those taxa with elaborate spines would invest less in their shells. We found the opposite: spinose individuals had shells that were 25% thicker. Thickened shells are frequently observed in response to shell peeling or crushing predators [9,34,42]. In addition, shell thickening can be induced by the presence of durophagous crabs or fluids from crushed conspecifics [43]. A hypothesis that may be deserving of attention is that brackish water should have higher levels of calcium carbonate, thereby stimulating taxa in the lower reaches of creeks to develop thicker shells at relatively low cost. A thicker shell in combination with elaborate spines would form a formidable defense, although we did not test this directly.

We assumed that defense has driven the elaboration of spines on the shells of the *Clithon* spp., however, we acknowledge that there may be alternate explanations. An intriguing hypothesis was proposed by Schilthuizen [44]. He suggested that shell ornamentation may be in response to sexual selection—at least in land snails. His hypothesis relies on the use of tactile cues during mating, in which snails detect the shell ornamentation of their mates. A role for sexual selection in spine formation has been proposed for other aquatic invertebrates [45], although direct evidence is lacking. It is possible that the spinose and nonspinose morphs of *Clithon prichardi* reflect sexual dimorphism, although this has never been examined.

In conclusion, the outcomes of the predictions we tested were consistent with spines playing a defensive role in the taxa we examined. Spine-bearing species dominated the entrance to the creek and were smaller in size. Levels of shell damage were low overall, with 2 of the 3 spinose taxa exhibiting no shell damage, as did many of the nonspinose taxa. Finally, shells of spinose species were 25% thicker, demonstrating increased rather than decreased investment. Conclusive evidence of a defensive role for spines awaits experimental manipulation of shell architecture in the presence of predators as has been undertaken for several molluscan taxa [7–9]. Nevertheless, this strikingly diverse, Indo-west Pacific fauna provides an opportunity to test a series of hypotheses and add to the generality of predator-prey interactions in an unusual system that has received limited attention. Unfortunately, these systems are also under threat from anthropogenic activities—changes in land use in particular [46]. This fauna offers rich prospects for future research.

**Author Contributions:** Conceptualization, A.R.D.; methodology, A.R.D., B.R. and A.H.; formal analysis, A.R.D. and M.J.R.; investigation, A.R.D., B.R. and A.H.; writing—original draft preparation, A.R.D.; writing—review and editing, all authors; funding acquisition, A.R.D. All authors have read and agreed to the published version of the manuscript.

**Funding:** This research received no external funding.

**Acknowledgments:** This project was made possible by support from the Centre for Sustainable Ecosystem Solutions, University of Wollongong. We are indebted to staff at the University of the South Pacific, particularly the late Bill Aalbersberg for facilitating this project. For assistance in the laboratory, we thank Kehani Manson, Allison Broad, Claudia Comacchio and Jose Abrantes. On Ovalau, we wish to thank Bobo and Karen Anger for their assistance and hospitality. This represents contribution number 323 from the Ecology and Genetics Group at the University of Wollongong. After more than 30 years contributing to our understanding of freshwater biodiversity in the South Pacific, Alison Haynes passed away before this manuscript was completed. We wish to dedicate it to her memory.

**Conflicts of Interest:** The authors declare no conflict of interest.

## Appendix A. Nonmarine molluscan fauna of Rukuruku Creek, Ovalau Island, Fiji, July 2014

*THIARIDAE*
**Melanoides** Olivier, 1804
*M. tuberculata* (O.F. Müller, 1774)

**Stenomelania** Fischer, 1885
*S. aspirans* (Hinds, 1844)
*S. persulcata* (Mousson,1869)

**Mieniplotia** Low and Tan 2014
*M. scabra* (O.F. Müller, 1774)

*NERITIDAE*
**Clithon** Montfort, 1810
*C. corona* (Linnaeus, 1758)
*C. diadema* (Récluz, 1841)
*C. olivaceum* (Récluz, 1843)
*C. pritchardi* (Dohrn, 1861)

**Neripteron** Lesson, 1831
*N. auriculatum* (Lamark, 1816)

**Neritina** Lamarck, 1816
*N. canalis* G.B. Sowerby I, 1825
*N. petitii* (Récluz, 1841)
*N. porcata* (Gould, 1847)
*N. pulligera* (Linnaeus, 1767)

**Neritona** von Martens, 1869
*N. macgillivrayi* (Reeve, 1855)

**Septaria** Férussac, 1807)
*S. bougainvillei* (Récluz, 1842)
*S. livida* (Reeve, 1856) (wide and narrow forms)
*S. freycineti* (Récluz, 1842)—synonym of *S. suffreni*
*S. macrocephala* (Le Guillou in Récluz, 1841)
*S. sanguisuga* (Reeve, 1856)

Identifications confirmed by Alison Haynes and names updated using WoRMS (World Register of Marine Species) with assistance from W.F. Ponder (Australian Museum).

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
