# Peer review of "Are Spine-Bearing Freshwater Gastropods Better Defended?"

_2673-4133, doi:10.3390/ecologies1010002_

Round 1
Reviewer 1 Report
Comments on Davis et al. “Are spine-bearing freshwater gastropods better defended?”
The authors present an interesting and valuable study of the occurrence of spines on neritid gastropod shells along a freshwater stream in Fiji in the context of protection against shell crushing predators and their distribution. The study appears straight-forward and I have only a few remarks of minor importance.
1) I am a non-native speaker but feel that the term protection fits better than defense.
2) There is an important study on a freshwater snail the authors have overlooked: Holomuzki JR & Biggs BJF 2006. Habitat-specific variation and performance trade-offs in shell armature of New Zealand mudsnails. Ecology 87: 1038-1047.
3) line 42: something is missing in this sentence
4) I see why it is probably not necessary, but I suggest to write a sentence in defense against the notion to eventually consider relationship in the statistical analysis. After all, all spiny species belong to a single genus.
5) is anything known about the potential role of phenotypic plasticity in C. pritchardi when it comes to spine formation? Is there some kind of induction through the presence of predators as assumed in Potamopyrgus antipodarum?
6) Why was the genus Septaria not considered?
7) Figure 2: lower and upper cases do not correspond in figure and caption.
8) line 157: 0.12 of what? 12% probably?
9) Table 2: the alignment of columns has gone astray; what about Neritina porcata and Neritona macgillvrayi? They are not listed here in contrast to Table 1.
10) Figure 5: I do not quite understand B and D. The horizontal line appears to be the mean shell thickness of a shell with mean length. Why is there dispersion along the line? I would expect the points in a single column if the X-axis is only presence/absence of spines. If this is simply for clarity, please, state so in the caption. The caption is too short anyway and in the very beginning something is missing. Relationship between shell thickness and what else?
11) line 223: are you sure that hollow spines break easier than massive ones?
12) It seems appropriate to point out that this study represents a single stream. Comparative analyses would certainly be more than welcome.
13) the author contributions mention funding acquisition and below follows the statement that there was not external funding.
14) line 278: typo in Bill Aalbersberg’s name
These are, as already mentioned, all minor issues. It should not be a big effort to address them appropriately and I am looking forward to seeing this account published in Ecologies in due course.
Author Response
We present the point by point comments of the reviewer and our responses in italics.
The authors present an interesting and valuable study of the occurrence of spines on neritid gastropod shells along a freshwater stream in Fiji in the context of protection against shell crushing predators and their distribution. The study appears straight-forward and I have only a few remarks of minor importance.
We thank the reviewer for their words of encouragement and their very thorough review.
1) I am a non-native speaker but feel that the term protection fits better than defense.
We assume that the reviewer is referring to our title. A quick search of the literature confirmed that 'spines' frequently appears with 'defense' in the titles of ecological publications, whereas 'spines' and 'protection' rarely appear together. Our preference is to continue using 'defense'
2) There is an important study on a freshwater snail the authors have overlooked: Holomuzki JR & Biggs BJF 2006. Habitat-specific variation and performance trade-offs in shell armature of New Zealand mudsnails. Ecology 87: 1038-1047.
This is indeed an important study and we thank the reviewer for bringing it to our attention. We now make brief reference to this publication in the text in relation to trade-offs.
3) line 42: something is missing in this sentence.
This sentence has been modified in keeping with the suggestion for simplifying it from Reviewer 2
4) I see why it is probably not necessary, but I suggest to write a sentence in defense against the notion to eventually consider relationship in the statistical analysis. After all, all spiny species belong to a single genus.
The opening paragraph in our Discussion and our first two tables highlight that only Clithon spp. bear spines. We feel that to restate this would simply be too repetitive.
5) is anything known about the potential role of phenotypic plasticity in C. pritchardi when it comes to spine formation? Is there some kind of induction through the presence of predators as assumed in Potamopyrgus antipodarum?
Part of the purpose of this contribution is to introduce scientists with an interest in defence to this striking fauna. Many unanswered questions remain and the role of predators in the induction of spine formation would be an intriguing question to answer. It is beyond the scope of the current contribution.
6) Why was the genus Septaria not considered?
This genus has a limpet like morphology and usually occurs in a distinct habitat - fast flowing sections of the stream. We considered the morphology very distinct from the Neritina and Clithon spp. on which we have focused and it's habitat of choice means it is less likely to encounter the predators of the other nerite taxa
7) Figure 2: lower and upper cases do not correspond in figure and caption.
Thank you - now modified.
8) line 157: 0.12 of what? 12% probably?
We see the confusion and although Figure 4 uses a proportion, we have elected to report in the text levels of shell damage as "rarely more than 12% of a sample from any location"
9) Table 2: the alignment of columns has gone astray; what about Neritina porcata and Neritona macgillvrayi? They are not listed here in contrast to Table 1.
The alignment of columns has now been fixed. Sample sizes for Neritina porcata and Neritona macgillvrayi were deemed too small for inclusion in Table 2. We now note this in the caption of Table 2.
10) Figure 5: I do not quite understand B and D. The horizontal line appears to be the mean shell thickness of a shell with mean length. Why is there dispersion along the line? I would expect the points in a single column if the X-axis is only presence/absence of spines. If this is simply for clarity, please, state so in the caption. The caption is too short anyway and in the very beginning something is missing. Relationship between shell thickness and what else?
The reviewer is correct. We have added "and shell length" to this caption. It now starts with "Relationships between shell thickness and shell length". The line in panels B and D are indeed mean shell length and the dispersion of points around the line are the partial residuals from the linear mixed model. We have now added "Partial residuals from the linear mixed models are presented in B and D" to the figure caption.
11) line 223: are you sure that hollow spines break easier than massive ones?
As we have not tested this directed and so we have opted to modify this sentence by adding the word "likely". It now reads "Their hollow nature likely renders them more prone to breakage"
12) It seems appropriate to point out that this study represents a single stream. Comparative analyses would certainly be more than welcome.
We have added a sentence to the Methods section which reads: "We contend that the fauna of Rukuruku Creek is typical of other creeks on the high islands of the Fijian archipelago, with many of the taxa we examined widely distributed across the Indo West Pacific."
13) the author contributions mention funding acquisition and below follows the statement that there was not external funding.
Internal rather than external funding was obtained.
14) line 278: typo in Bill Aalbersberg’s name
Corrected thank you.
These are, as already mentioned, all minor issues. It should not be a big effort to address them appropriately and I am looking forward to seeing this account published in Ecologies in due course.
As a direct result of this reviewers input we have added this reference to the manuscript:
Holomuzki, J.R.; Biggs B.J.F. Habitat-specific variation and performance trade-offs in shell armature of New Zealand mudsnails. Ecology 2006, 87, 1038-1047.
Reviewer 2 Report
General comments:
The authors test whether spines on the shells of nerite snails help protect them from predation. Some interesting, suggestive data are presented, but I have three major concerns with the present manuscript.
1) The authors show that the incidence of spine-bearing snails increases toward the mouth of the study stream where it is said that predation is highest. This is an interesting finding, but the authors provide no estimates of predation pressure along the length of the stream. Do the authors have longitudinal data on frequency of shell damage? Table 2 shows that some snails had zero damage both at the stream mouth and at its highest point. In addition, the frequency of spine-bearing individuals shows no consistent relationship with stream location in the dimorphic Clithon pritchardi. Do the authors have data on the longitudinal abundance and distribution of snail predators in the study stream? Hardly anything is said about the potential predators living in the study stream.
2) The authors’ hypothesis that spines help protect against predation is contradicted by the lack of a strong association between spine-bearing and shell damage (see Table 2). The most controlled comparison is between spinose and non-spinose individuals of Clithon pritchardi, which do not differ in shell damage.
3) The authors do not consider whether other forms of selection besides that related to predation may be affecting spine-bearing in nerite snails. For example, sexual selection may play an important role, especially in C. pritchardi, which includes both spinose and non-spinose forms. Have these forms been sexed? As in the sexually dimorphic isopod Deto echinata (Glazier et al. 2016), the spines of this species are blunt and somewhat curved (not sharply pointed), and thus are not likely effective at puncturing the skin of potential predators. However, they could still be effective at making ingestion difficult for small-gaped predators. Have the authors ever observed predation on their study snails? It is possible that the spines may be sexually dimorphic displays, at least in C. pritchardi. It would be interesting and worthwhile to consider this possibility.
Specific comments:
L 23: Change “exhibited” to “exhibiting”?
L 38: To reduce wordiness, I suggest changing “act to encouraging” to “encourage”.
L 42: I suggest changing “reaches its most impressive” to “is most impressive”?
L 60-71, 106-107: Is there any evidence of sexual dimorphism of spine length in nerite snails, as observed in some other animals (e.g., see Glazier et al., 2016)? Is it possible that spinose individuals of C. pritchardi are males, whereas non-spinose individuals are females? If so, sexual selection may also be involved in the evolution of spines in these snails. Schilthuizen (2003) discusses the possible relevance of sexual selection to snail ornamentation.
L 78-79: Run-on sentence.
L 120-121: Please clarify.
L 166: Incomplete sentence.
Table 2 requires reformatting so that data are properly aligned.
L 244-246: Not a complete sentence.
Literature cited:
Glazier, D. S., Clusella‐Trullas, S., & Terblanche, J. S. (2016). Sexual dimorphism and physiological correlates of horn length in a South African isopod crustacean. Journal of Zoology 300, 99-110.
Schilthuizen, M. (2003). Sexual selection on land snail shell ornamentation: a hypothesis that may explain shell diversity. BMC Evolutionary Biology 3, 13.
Author Response
We found this reviewer raised some intriguing points - challenging out preconception that spines were purely defensive. We have copied their comments below with our responses in italics.
1) The authors show that the incidence of spine-bearing snails increases toward the mouth of the study stream where it is said that predation is highest. This is an interesting finding, but the authors provide no estimates of predation pressure along the length of the stream. Do the authors have longitudinal data on frequency of shell damage? Table 2 shows that some snails had zero damage both at the stream mouth and at its highest point. In addition, the frequency of spine-bearing individuals shows no consistent relationship with stream location in the dimorphic Clithon pritchardi. Do the authors have data on the longitudinal abundance and distribution of snail predators in the study stream? Hardly anything is said about the potential predators living in the study stream.
As we state, our overarching hypothesis was that spines play a defensive role and we developed a series of predictions in support of this hypothesis. The reviewer is correct in that we do not present estimates of predation pressure other than it being high at the creek entrance and we then assume it decreases with distance from the ocean. We do present data on the frequency of shell damage in Fig. 4. We have also sought the opinion of three ichthyologists working in the South Pacific regarding predators in the upper and mid reaches of streams. We have added the following text to the Discussion:
"Predators are not restricted to creek entrances, however. The mid and upper reaches of freshwater systems in Fiji are frequented by several invertebrate feeders including eels Anguilla spp., the goby, Awaous guamensis, the gudgeon, Eleotris fusca, and several representatives of the Kuhliidae (David Boseto, Ecological Solutions, Solomon Islands, pers. comm.) [37]."
2) The authors’ hypothesis that spines help protect against predation is contradicted by the lack of a strong association between spine-bearing and shell damage (see Table 2). The most controlled comparison is between spinose and non-spinose individuals of Clithon pritchardi, which do not differ in shell damage.
We agree that shell damage and spine bearing represents a weak association. Levels of shell damage in the marine environment can be strikingly high - more than 90% for some species in SE Australia (Coleman & Davis, unpublished data), while in the current study levels were very low. We indicate that not all of our predictions were borne out (noted in the Abstract and the first paragraph of the Discussion) and emphasise that relative levels of shell damage were low overall. The weight of evidence that we have presented supports the notion that spines play a defensive role.
3) The authors do not consider whether other forms of selection besides that related to predation may be affecting spine-bearing in nerite snails. For example, sexual selection may play an important role, especially in C. pritchardi, which includes both spinose and non-spinose forms. Have these forms been sexed? As in the sexually dimorphic isopod Deto echinata (Glazier et al. 2016), the spines of this species are blunt and somewhat curved (not sharply pointed), and thus are not likely effective at puncturing the skin of potential predators. However, they could still be effective at making ingestion difficult for small-gaped predators. Have the authors ever observed predation on their study snails? It is possible that the spines may be sexually dimorphic displays, at least in C. pritchardi. It would be interesting and worthwhile to consider this possibility.
This is an intriguing point, we thank the reviewer for mentioning it and directing us to this literature. We did not observe any acts of predation, but highlight that many, perhaps all, of these taxa were nocturnal and we did not have access to a lab. on Ovalau. We have now added a paragraph to the Discussion that reads:
"We have assumed that defence has driven the elabouration of spines on the shells of the Clithon spp. we have examined, but we acknowledge that there may be alternate explanations. An intriguing hypothesis has been proposed by Schilthuizen [44]. He suggested that shell ornamentation may be in response to sexual selection – at least in land snails. His hypothesis relies on the use of tactile cues during mating in which snails detect the shell ornamentation of their mates. A role for sexual selection in spine formation has been proposed for other aquatic invertebrates [45], although direct evidence is lacking. It is possible that the spinose and non-spinose morphs of Clithon prichardi reflect sexual dimorphism, although this has never been examined."
Specific comments:
L 23: Change “exhibited” to “exhibiting”?
Done
L 38: To reduce wordiness, I suggest changing “act to encouraging” to “encourage”.
Done
L 42: I suggest changing “reaches its most impressive” to “is most impressive”?
Done
L 60-71, 106-107: Is there any evidence of sexual dimorphism of spine length in nerite snails, as observed in some other animals (e.g., see Glazier et al., 2016)? Is it possible that spinose individuals of C. pritchardi are males, whereas non-spinose individuals are females? If so, sexual selection may also be involved in the evolution of spines in these snails. Schilthuizen (2003) discusses the possible relevance of sexual selection to snail ornamentation.
These are all intriguing possibilities and the late co-author Alison Haynes would have been well placed to address the question of whether spines are sex linked in C. pritchardi as she has dissected all of these taxa. The issue of sexual selection and snail ornamentation now appears as a paragraph in the Discussion as outlined in our answer to point 3) above.
L 78-79: Run-on sentence.
We have recast this sentence. It now reads "The topography at these lower stations was relatively flat, the creek shallow...and with flow rates of around 10cm per second."
L 120-121: Please clarify.
We agree that this was a clumsy sentence. We have now split and recast it. It now reads: "Our focus was on the members of the genera..... Clithon and Neritina were each represented by four species, while Neritona was represented by a single taxon."
L 166: Incomplete sentence.
Now joined to previous sentence with a semicolon.
Table 2 requires reformatting so that data are properly aligned.
Done - not sure how that was submitted with such poor alignment
L 244-246: Not a complete sentence.
Sentence now modified to read "Amphidromy is where adults are resident in freshwater while their larvae are swept downstream..."
As a direct result of this reviewers comments we have now cited the following:
Glazier, D. S.; Clusella‐Trullas, S.; Terblanche, J. S. Sexual dimorphism and physiological correlates of horn length in a South African isopod crustacean. J. Zool. 2016, 300, 99-110.
Schilthuizen, M. Sexual selection on land snail shell ornamentation: a hypothesis that may explain shell diversity. BMC Evol. Biol. 2003, 3, 13-19.
Copeland, L.K.; Boseto, D.T.; Jenkins, A.P. Freshwater ichthyofauna of the Pacific-Asia biodiversity transect (PABITRA) gateway in Viti Levu, Fiji. Pac. Conserv. Biol. 2016, 22, 236-241.
Round 2
Reviewer 2 Report
The authors have improved their manuscript somewhat, but I am still concerned that they have not explicitly mentioned the following points:
1) The gradient of increasing predation risk toward the mouth of the study creek is inferred, not directly measured. This should be stated clearly.
2) The authors state in their abstract and the first paragraph of their Discussion section that not all of their predictions were supported. What these predictions are should be explicitly stated in both of these places. This includes not only the prediction that spinose snails should have thinner shells (they don’t), but also that shell damage should be lower in spinose snails, which is not necessarily the case. The strongest evidence against this prediction is that spinose and non-spinose individuals of Clithon pritchardi do not differ in shell damage (as I mentioned in my original review). The authors have chosen not to mention this contrary evidence, but to avoid a biased presentation, I believe that it should be mentioned explicitly.
Overall, the authors' evidence that spines are defensive in their study snails is suggestive, not conclusive.
Author Response
Apologies - we did not realise that the reviewer was seeking explicit statements in the manuscript. The Reviewer raised two points:
1) The gradient of increasing predation risk toward the mouth of the study creek is inferred, not directly measured. This should be stated clearly.
We have now added an explicit sentence to this effect in the Discussion: "We wish to emphasise though that we have inferred an elevated predation risk at the creek entrance, but it was not quantified."
2) The authors state in their abstract and the first paragraph of their Discussion section that not all of their predictions were supported. What these predictions are should be explicitly stated in both of these places.
Our Abstract states outcomes of each of the four predictions and we have now removed "we conclude" and modified the final sentence in the Abstract to read: "Taken together these findings suggest that the elaborate spines of Clithon spp. play a defensive role."
The concluding paragraph in the Discussion now states each of the predictions we examined: "In conclusion, the outcomes of the predictions we tested were consistent with spines playing a defensive role in the taxa we examined. Spine-bearing species dominated the entrance to the creek and were smaller in size. Levels of shell damage were low overall, with 2 of the 3 spinose taxa exhibiting no shell damage, as did many of the non-spinose taxa. Finally, shells of spinose species were 25% thicker, demonstrating increased rather than decreased investment. Conclusive evidence of a defensive role for spines awaits experimental manipulation of shell architecture in the presence of predators as has been undertaken for several molluscan taxa [7-9]."
We trust that these changes answer the reviewers concerns.